# Assessment of Youth Water Polo Players’ Swimming Sprint Potential: A New Approach to Building an International Model

**DOI:** 10.3390/jfmk10020200

**Published:** 2025-05-31

**Authors:** Andrea Perazzetti, Antonio Tessitore, Mehmet Zeki Özkol, Nebojša Novoselac, Milivoj Dopsaj

**Affiliations:** 1Department of Movement, Human and Health Sciences, University of Rome ‘Foro Italico’, 00135 Rome, Italy; antonio.tessitore@uniroma4.it; 2Faculty of Sport Sciences, Ege University, Izmir 35040, Türkiye; zeki.ozkol@gmail.com; 3German Swimming Association, 1886 Kassel, Germany; nebojsawpolo@gmail.com; 4Faculty of Sport and Physical Education, University of Belgrade, 11000 Belgrade, Serbia; milivoj.dopsaj@fsfv.bg.ac.rs

**Keywords:** front crawl, swimming styles, ball abilities, skill index, performance analysis, team sport

## Abstract

**Background**: To cope with their horizontal swimming phases, water polo players use different swimming techniques, such as specific variants of the crawl swimming style. Thus, this study aimed to investigate the swimming skills of young water polo players. **Methods**: An all-out 25-m sprint swimming test in crawl style was completed by 273 international youth water polo players (age = 14.0 ± 0.8 yrs) in two modalities: basic crawl with the head in the water (25C_HeadIN_), and a crawl performed while dribbling the ball (25C_Ball_). **Results**: We registered an average time of 14.79 and 15.64 s for 25C_HeadIN_ and 25C_Ball_, respectively, in which the ball dribbling increased to 5% of the swimming time. A swimming skill index (25C_SIC_) was calculated to account for differences in ball dribbling speeds, which, considering our international sample and in the absence of previous data, we could speculate as the first international standard value for 14-year-old male water polo players competing at international level. The averaged values for 25C_SI_ and 25C_SIC_ were 0.94 ± 0.04 (a.u.) and 1.52 ± 0.15 (a.u.), respectively. Factor analysis indicated that swimming with and without the ball are structurally distinct technical skills, highlighting the specificity of these water polo players’ abilities. Moreover, the study shows significant differences (*p* < 0.05) between players from different countries and despite some limitations, its results provide valuable insights for the assessment and development of sprint swimming skills in youth water polo players. **Conclusions**: In summary, the findings of this research provide practical implications for training, player selection, player development and the optimization of youth water polo player performance.

## 1. Introduction

Water polo is an aquatic team sport requiring a specific set of skills that each player should utilize according to their playing position or game situations [1,2]. Modern water polo is characterized by rapid counterattacks, frequent sequences of passes leading to shots, as well as zone or pressing systems of play that emphasize the intensity of the actions (both defensive and offensive), accentuating active contact with opponents [3,4]. The scientific literature and the ongoing rule changes [5] demonstrate that this discipline requires players to maintain an exceptionally high level of speed, strength, and repeated sprint ability through the match [6,7,8]. Alongside the development of water polo’s rules and structure, in recent years, the training process of young players has been improved and adapted in relation to the needs of the elite senior players [9,10,11,12]. Consequently, nowadays there is a growing need for the development of a specific youth water polo methodology, encompassing talent selection, training, testing, and monitoring the effects of the training processes [13,14,15,16].

Beyond technical and tactical skills, swimming abilities play a crucial role in water polo performance. According to the review by Botonis et al. [17], the total distance covered in international competitions (for both male and female players) is more than 1000 m per match, with varying swimming velocities [18]. Most of the time, these swimming actions last less than 20 s and are characterized by very intense horizontal movements [19]. Specifically, the crawl (or freestyle) stroke or its variants are considered to be the techniques most frequently performed by water polo players, with an average duration of 12 s per action [17]. When performing these swimming techniques, players apply the crawl swimming style in various modalities, both with and without the ball. They can crawl with their head above the water, crawl while dribbling or holding the ball, or use a combination of crawl techniques, such as arm work in front crawl and leg work in breaststroke [20,21]. Although many studies have analyzed the swimming tests most frequently used to evaluate water polo players’ abilities [22] or highlight the importance of using the ball during swimming practice [23,24], there is still a lack of research providing evidence-based tools for monitoring and improving the training progress of young water polo players. As demonstrated in sport literature, team sport coaches, especially those working with younger players, need these kinds of tools to monitor their players’ improvements without interrupting the training process, ensuring that test results are both accurate and comparable to established standards for the relevant age demographic [14]. For this reason, we state that it is necessary to develop a simple and scientifically approved field method that allows water polo coaches to test their players and compare the obtained results with a validated standard scale.

Therefore, in this study based on youth international players from different national European water polo schools [16], our main aim was to assess the participants’ crawl-style swimming performance in both basic and specific modalities (i.e., those informed by the position of the head and use of the ball) during a 25-m sprint test. Through the statistical analysis of the results, the final scope of this work will be to provide an evidence-based statistical tool that will be useful for evaluating water polo players’ swimming abilities.

## 2. Materials and Methods

A field test protocol was applied to collect experimental data on the sprint swimming ability of young water polo players. The research was realized in accordance with the Helsinki Declaration for Recommendations Guided by Physicians in Biomedical Research Involving Humans and was approved by the Ethical Committee of the Faculty of Sports and Physical Education at the University of Belgrade (document number 484–2). We informed all players and their clubs, national federations, and parents about the purpose and goal of the tests. All players participated in the study after at least one of their parents signed a tailored consent form.

### 2.1. Subjects

Two hundred and seventy-three (*n* = 273) youth water polo players (age = 14.0 ± 0.8 yrs., BH = 174.0 ± 9.0 cm, BW = 64.6 ± 11.3 kg, BMI = 21.23 ± 2.87 kg/m², training experience = 5.7 ± 1.3 yrs.) from Serbia (*n* = 104), Slovenia (*n* = 11), Türkiye (*n* = 15), Italy (*n* = 41), and Germany (*n* = 102) were recruited to be assessed in this study. Players from Italy belonged to a youth elite First League club (U14 of the S.S. Lazio Nuoto), while participants from other countries were recruited from their youth national teams. In particular, players from Slovenia and Türkiye belonged to the selection of youth national teams before international competitions, while players from Serbia and Germany were recruited during their national camps (organized for the selection process). In this way, all participants can be classified as selected young players competing at international level. All tests were submitted during training sessions/camps from the 2022/23 water polo season.

### 2.2. Testing Procedures and Data Collection

All subjects were assessed by performing an all-out 25-m sprint test swum in crawl style with two different modalities [21]: (1) basic crawl swimming with the head in the water (25C_HeadIN_), in which players performed a crawl executed with the typical face-down position assumed by swimmers; and (2) crawl swimming while dribbling the ball (25C_Ball_), in which players performed a crawl executed with the head raised from the water, which is typical of water polo players. Both 25C_HeadIN_ and 25C_Ball_ tests were carried out in a 25m pool. For the 25C_Ball_ test, players used a “Mikasa” water polo size 5 ball (Mikasa Corporation, Hiroshima, Japan).

Before the commencement of the tests, all participants executed a standardized warm-up protocol, which included all swimming styles and specific water polo exercises performed for a total of 800 m, followed by 5 min of passive rest. To ensure the standardization of the test, the following aspects were controlled: (a) use of a standardized starting position with their feet in contact with the wall (Figure 1); (b) players started when an acoustic signal was given by a starter, who simultaneously activated the measurement of the completion time; (c) the test was stopped as soon as the player touched the end wall with one hand.

All trials were recorded by means of a video camera (Sony FDR-AX43; Sony, Tokyo, Japan), while the measurement of completion time was provided using a digital quartz stopwatch with a measurement that was precise to the second decimal place (Finis, 3X300M, Tracy, CA, USA). The trial footage was later used to analyze the crawl executions according to the two protocols’ modalities by reproducing them using the Kinovea software (release version 9.5).

All individual tests were performed twice using a randomized procedure, with a 3-min rest between attempts, while the best time was selected for the statistical analysis.

Furthermore, the data were used to provide two swimming skill indexes, calculated as follows: (1) the crawl skill index (25C_SI_), calculated as the ratio between the test with basic crawl swimming and crawl swimming while dribbling the ball (25C_HeadIN_/25C_Ball_) and expressed in percentage; and (2) the corrected crawl skill index (25C_SIC_), calculated by the formula 25/25C_Ball_ x 25C_SI_, and expressed in arbitrary units.

### 2.3. Statistical Analysis

A descriptive statistic (mean, SD, cV%, Min and Max, Std. Error) of tests and indexes were provided. The absolute and relative error of the arithmetic mean was calculated to indicate the measurement precision and the 95% confidence interval (lower and upper bounds) for the mean was provided.

An exploratory factor analysis model, with Varimax rotation, was used to define the structure of the measured space. Then, a multidimensional scaling method was used to define centroid scores to define the position of each player in the sample for each extracted factor [25]. In this way, each individual player was positioned according to their swimming ability within an overall sample group of young water polo players. In the following procedure, all defined centroid values were transformed into a proportional numerical scale from 0 (hypothetical minimum) to 100 (maximum) points, in accordance with the previously described procedure [25,26]. Finally, multiple regression analysis was applied to define the three age groups and sports-specific equations of the model of the sprint and sprint skill swimming potential of the young water polo players, as well as the CSS_P_ (crawl sprint swimming potential (first factor score for 25C_HeadIN_ and 25C_Ball_)); CSS_SP_ (crawl sprint swimming skill potential (second factor score for 25C_SI_ and 25C_SIC_)); and OCSS_S_ (overall sprint crawl swimming score (25C_HeadIN_ and 25C_SI_ as the most representative variables of the factors)). The differences between the scores of respondents from different countries were determined using One-Way ANOVA, and post hoc multiple comparison was conducted according to Bonferroni’s criteria to determine the sensitivity of the applied models. All analyses were performed by the statistical software package SPSS (IBM, SPSS Statistics, version 29), and the level of statistical significance was defined by 95% and the probability values of *p* < 0.05 [27].

## 3. Results

The results of the descriptive statistics are shown in Table 1.

The results of the coefficient of variation (Table 1) shows that all swimming variables were very homogeneous (the cV% was between 4.54 for 25C_SI_ and 10.18 for 25C_SIC_). Moreover, the measurement procedure itself was very consistent between the measurers, because the standard error of the arithmetic mean of the results (relative values) was at a level of less than 1% (in Table 1, the Std. Error. was between 0.32% for 25C_SI_ and 0.59% for 25C_SIC_). The Kaiser–Meyer–Olkin measure of sampling adequacy (KMO) value was determined to be 0.396, and Bartlett’s Test of Sphericity resulted in a value of 2797.2, both with a statistically significant level of *p* < 0.001, which indicates the statistically significant adequacy of data for multivariate statistical analysis.

Table 2 shows the results of the factor analysis, in which we extracted two components (factors) with very proportionally explained variances at rotation matrixes of 54.67 and 45.08% (i.e., a cumulative value of 99.75%). In the rotated component matrix (Table 2), the first extracted factor is saturated with two variables, i.e., 25C_HeadIN_ and 25C_Ball_, while the second is saturated with the remaining two variables, i.e., 25C_SI_ and 25C_SIC_.

The defined models for evaluating the crawl sprint swimming potential (CSS_P_), crawl sprint swimming skill potential (CSS_SP_), and overall sprint crawl swimming score (OCSS_S_) in young water polo players are represented by the following Equations:CSS_P_ = 280.0319 − (25C_HeadIN_ × 8.0457) − (25C_Ball_ × 7.0992)CSS_SP_ = −237.3686 + (25C_SI_ × 210.6363) + (25C_SIC_ × 57.7609)OCSS_S_ = 437.0765 − (25C_HeadIN_ × 9.9386) − (25C_SI_ × 253.7106)

Table 3 presents the descriptive statistics for swimming scores across different subgroup countries and the results of comparative statistics highlighting the differences between them.

## 4. Discussion

The study’s results show how the mean completion times of the fourteen-year-old young male water polo players in the all-out 25-m sprint test differed based on the performance modality of two different crawl executions. The completion time of the 25C_HeadIN_ test, performed with the basic crawl technique, showed a range between 12.48 (min) and 18.68 (max) seconds, while the 25C_Ball_ test ranged between 12.67 (min) and 21.09 (max) seconds. These ranges can be explained by the players’ swimming techniques and ball-dribbling ability [24], as well as their biological age [28,29]. Based on published data on young water polo players of the same age from Serbia [21], our results show a worse performance compared to previous youth national teams (25C_HeadIN_ = 13.92 ± 0.82; 25C_Ball_ = 15.00 ± 1.13 s) and a better performance compared to youth players from clubs (25C_HeadIN_ = 17.30 ± 2.50; 25C_Ball_ = 19.41 ± 3.91 s). Considering studies on players’ positions, our data are consistent with those published in previous research with athletes of the same age. A study by Chaplins’kyy et al. [30] reported that the average time taken to complete the 25C_HeadIN_ was 14.45 ± 0.18, 14.15 ± 0.10, and 14.15 ± 0.30 s for attackers, defenders, and midline players, respectively, confirming the external validity of our results and arguing that the obtained models can find valid application in youth water polo practice.

Comparing the mean completion time of the two all-out 25-m tests submitted in our study, the swimming time with the ball (25C_Ball_) was 5.3% higher than the swimming time using the basic crawl technique (25C_HeadIN_). Considering the crawl skill index value (25C_SI_ = 0.947), it can be argued that the extra time needed to complete the 25C_Ball_ is due to changes in the head and body position, as well as the need to control the ball while swimming. (Table 1). However, it is possible for two players to have the same level of ball handling skill (i.e., the same nominal 25C_SI_ value) but swim with the ball at different speeds. To address this issue, we introduced a new corrected swimming skill index (25C_SIC_) to our study. It gave us a mean value of 1.523 ± 0.155, which, until more data from different international samples can be obtained, can be considered as an initial international standard value for 14-year-old male water polo players (Table 1). Factor analysis confirmed that the ability to swim at maximal intensity over 25 m is fundamentally different from the ability to swim the same distance while dribbling the ball. For this reason, they can be considered as two distinct skills, each belonging to a separate factor space (Table 2). Indeed, the most discriminating tests were the ability to swim at maximal intensity using the front crawl technique over 25 m (25C_HeadIN_) (which had a factor saturation of 0.990) and the basic index skill of swimming with a water polo ball (25C_SI_) (which had a factor saturation of 0.998). In a previous study by Uljević et al. [10], which used a similar methodology involving a sample of 54 young male water polo players (ranging from 15 to 17 years of age) from three top-level Croatian teams, the factor analysis determined three independent specific motor latent dimensions: shooting capacity, jumping capacity, and sprint swimming capacity. The authors concluded that this finding clearly shows that players’ abilities in the water are highly specific, and this conclusion was supported by our data, which demonstrated that these skills also have a sub-specific nature. This means that simple movements in water that appear motorically similar are structurally different from the specific skills required when playing water polo [31]. Considering that the sample of our study consisted of highly trained young water polo players (competing at international level) from five different countries, these results can be considered highly representative for an initial generalization of the investigated phenomenon. Furthermore, the significant between-group difference found in this study for all indexes, with the largest difference registered for the OCSSs model (Table 3, OCSSs, F = 16.936, *p* < 0.001), confirm the sensitivity of these indexes, which are able to detect differences in scores that can be explained by the influence of the different national water polo schools attended by the players [16]. In relation to the between-group differences, the lowest score (and higher standard deviation) registered for the Italian sample, particularly for CSS_P_ (Table 3, 39.40 ± 21.77), can be explained by the fact that these players belonged to a youth elite first league club (S.S. Lazio Nuoto), compared to the national team players from other countries. However, this fact can be positively interpreted, showing the variability in swimming abilities typical of youth club players.

Additionally, the equation used to calculate the CSS_P_, CSS_SP_, and OCSS_S_ models allowed us to standardize outcomes that encompass a wide range of values. In fact, the sample included low scores (−19.98, 8.76, and 1.28 for CSS_P_, CSS_SP_, and OCSS_S_, respectively, Table 1) as well as high scores (106.39, 100.66, and 92.13 for CSS_P_, CSS_SP_, and OCSS_S_, respectively, Table 1). These values reflect the full performance range expected in young water polo players, further supporting the robustness of the indexes provided in this study. Indeed, the equation models obtained in this study indicate the high practical applicability of the use of the 25C_HeadIN_ and 25C_Ball_ tests and skill index with youth male water polo players to assess their individual swimming speed in an all-out test over a short distance with different water polo skill modalities. The approach provided in our study allows club and national team head coaches to assess the sprinting ability of young water polo players, identifying any weaknesses in sprint development using applied training technology and tailored field tests. The findings of this study can be used for systematic and structured selection of teams and players, determining the appropriate match concept, and organizing specific training sessions.

## 5. Conclusions

Researchers have conducted most studies in this field under highly controlled conditions [22,23]. While such settings are essential for obtaining precise and reliable data, it is equally important to conduct research in real-world scenarios to ensure ecological validity and practical applicability [32]. The new international water polo rules, which adopt a 25-m playing field, make this study highly specific and relevant to the development game format of water polo [33]. Indeed, the equation models from this study could serve as a valid innovative tool for assessing the training process in a specific age group (e.g., 14-year-old males) according to players’ individual sprint swimming ability. In this way, the training of young water polo players can partly be realized according to the strictly controlled effects of their development (i.e., as a deterministic model of the training process). Moreover, the scores from the two all-out tests in this study (25C_HeadIN_ and 25C_Ball_), which show the players’ potential skills in sprint swimming, can be very helpful for assessing their individual speeds. However, our study has some limitations, including the fact that our sample comprised players of different performance levels (top-tier national teams, second-level national teams, and a single club team) and was restricted to only five countries, which, despite some of them having a compelling and successful tradition in this sport discipline, could not reflect the level of countries with a less widespread water polo tradition. Our study focused on a specific sex and age group (males, 14 years old), which should be expanded to include other youth categories and should involve female players to provide a more comprehensive understanding of youth water polo across ages and both sexes. Additionally, for the youth categories around the peak height velocity, the analysis should also include an assessment of players’ biological maturation, which could influence the results.

Future studies on this topic could also investigate other swimming demands beyond 25-m sprints, such as change of direction, shorter distance or different starting conditions (i.e., without pushing away from the wall) to further validate such specific indexes for the sprint swimming skills of young water polo players.

## Figures and Tables

**Figure 1 jfmk-10-00200-f001:**
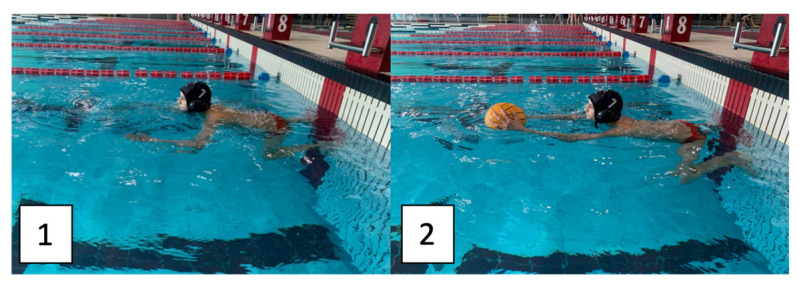
Starting position involving pushing from the wall, without the ball (**1**) and with the ball (**2**). Note. This photo was previously used during an international conference presentation of the authors [24], and the player’s parents signed their consent to the publication of this photo.

**Table 1 jfmk-10-00200-t001:** Descriptive statistics of all swimming tests and indexes.

		25C_HeadIN_	25C_Ball_	25C_SI_	25C_SIC_	CSS_P_	CSS_SP_	OCSS_S_
Mean		14.79	15.64	0.947	1.523	50.00	50.00	50.00
SD		1.08	1.23	0.043	0.155	16.67	16.67	16.67
cV%		7.30	7.86	4.54	10.18	33.34	33.34	33.34
Min		12.48	12.67	0.829	1.050	−19.98	8.76	1.28
Max		18.68	21.09	1.072	1.528	106.39	100.66	92.13
Std. Error. (aps, sec).		0.066	0.074	0.003	0.009	1.009	1.009	1.009
Std. Error. (rel, %).		0.45	0.47	0.32	0.59	2.02	2.02	2.02
95% Confidence Interval for Mean	Lower Bound	14.66	15.50	0.941	1.505	48.01	48.01	48.01
	Upper Bound	14.92	15.79	0.952	1.542	51.99	51.99	51.99

Note. 25C_HeadIN_: crawl swimming with head in the water; 25C_Ball_: crawl swimming while dribbling the ball; 25C_SI_: crawl skill index; 25C_SIC_: corrected crawl skill index; CSS_P_: crawl sprint swimming potential; CSS_SP_: crawl sprint swimming skill potential; OCSS_S_: overall sprint crawl swimming score.

**Table 2 jfmk-10-00200-t002:** Factor analysis: total variance explained and rotated component matrix.

**Total Variance Explained Matrix**						
Component	Initial Eigenvalues			Rotation Sums of Squared Loadings		
	Total	% of Variance	Cumulative %	Total	% of Variance	Cumulative %
1	2.704	67.602	67.602	2.187	54.668	54.668
2	1.286	32.149	99.751	1.803	45.082	99.751
3	0.009	0.224	99.975			
4	0.001	0.025	100.00			
**Rotated Component Matrix**						
	Component					
	1			2		
25C_HeadIN_	**0.990**			0.142		
25C_Ball_	**0.898**			−0.436		
25C_SI_	0.056			**0.998**		
25C_SIC_	−0.631			**0.773**		

Note. 25C_HeadIN_: crawl swimming with head in the water; 25C_Ball_: crawl swimming while dribbling the ball; 25C_SI_: crawl skill index; 25C_SIC_: corrected crawl skill index.

**Table 3 jfmk-10-00200-t003:** Descriptive statistics and post hoc analysis of the swimming score models between countries.

Subgroup	CSS_P_	Subgroup	CSS_SP_	Subgroup	OCSS_S_
Serbia *	57.95 ± 13.99	Germany #	55.81 ± 15.28	Serbia †	57.58 ± 16.49
Slovenia	54.35 ± 12.62	Serbia #	50.27 ± 16.73	Slovenia	57.29 ± 10.43
Germany	46.67 ± 14.16	Slovenia	46.47 ± 10.07	Türkiye	56.22 ± 14.65
Türkiye	43.36 ± 10.04	Italy	40.89 ± 16.23	Italy	48.63 ± 14.16
Italy *	39.40 ± 21.77	Türkiye	36.09 ± 10.92	Germany †	41.13 ± 14.14
ANOVA	F = 14.17 *p* < 0.001		F = 10.104 *p* < 0.001		F = 16.936 *p* < 0.001
* Serbia vs. Germany, Türkiye and Italy, *p* < 0.001, =0.006 and <0.001, respectively. * Italy vs. Slovenia, *p* = 0.042. # Germany vs. Italy and Türkiye, *p* < 0.001 & <0.001, respectively. # Serbia vs. Italy and Türkiye, *p* = 0.013 and 0.012, respectively. † Serbia vs. Italy *p* = 0.014 † Germany vs. Serbia, Türkiye and Slovenia, *p* < 0.001, =0.003 and =0.008, respectively.					

Note. CSS_P_: crawl sprint swimming potential; CSS_SP_: crawl sprint swimming skill potential; OCSS_S_: overall sprint crawl swimming score.

## Data Availability

Data are contained within the article.

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
