# Peer review of "Assessment of Youth Water Polo Players’ Swimming Sprint Potential: A New Approach to Building an International Model"

_jfmk, 2025, doi:10.3390/jfmk10020200_

Round 1
Reviewer 1 Report
Comments and Suggestions for Authors
Manuscript needs detailed proofreading.
Comments:
Title: „Simple Factorial Modeling of Sprint Swimming Potential in Youth Water Polo Players: an International Data Approach“ is ambiguous.
Comment: What is „Simple Factorial Modeling“? International data approach - simply not appropriate phrase.
Summary: The participants were elite young water polo players (14.0 ± 0.8 years), and the 17
authors assessed their sprint swimming times in the basic front crawl technique over 25 18
meters, both with and without the ball.
Comment: Phrase „Authors Assessed“ is not needed.
Summary: Results: Results showed an average time of 14.79 seconds for the crawl technique and 15.64 seconds for the crawl while leading the ball, revealing that leading the ball increase swim time by approximately 5%.
Comment: Not uderstandable and data confuses reader without konowledge of tests. This have to be rewritten
The study introduced a swimming skill index (25CSIC) to account for differences in leading-ball speeds, which was established as a potential international standard for 14 years old players
Comment: „potential international standard“ authors can think so, but it have to be real
Line 88 Players from Italy belonged to a youth elite First League club (S.S. Lazio Nuoto), while participants from other countries were recruited from their youth national teams
Comment: Youth national teams vs first league club – seems unappropriate
Line 123: expressed in arbitrary unit.
Comment: what does that means? Arbitrary unit? Please clarify!
Lines 132-135 A descriptive statistic (mean, SD, cV%, Min and Max, Std. Error) of tests and indexes
were provided. Considering that this was a multicenter study, the error of the arithmetic mean (absolute and relative value) was calculated as the value of the measurement precision, as well as 95% lower and upper bound confidence interval for mean. In this way, all results were fully descriptively analyzed.
Comment: What is definition of multicenter study? If you have several data collection locations across globe that does not mean that it is a multicenter study. Phrase „the error of the arithmetic mean (absolute and relative value)“ is unclear. Results were not „descriptively analyzed“. Shortly, all paragraph have to be rewritten. Also, usage of FA for 4 variables is weak.
Line 137:
Factor analysis, exploratory model with Principal component analyses extraction and
with Varimax with Kaiser normalization rotation, was used to define the structure of the
measured space. Then, a multidimensional scaling method was used to define centroid 139
scores to define the position of each player in the sample for each extracted factor [25].
this way, each individual player is positioned according to defined swimming abilities
within a group of selected young water polo players from different countries, that is, in
an international specific sport-selected group. Mathematical procedures were used to
transform the integral centroid score value of each subject into a numerical analogy, mean
ing that was transformed into a proportional numerical distribution score on a linear scale,
where the value of centroid test score for each individual participant was transformed into
a proportional point score on a scale from 0 (hypothetical minimum) to 100 (hypothetical
maximum) points
Comment: this is unclearly written. For example: „…with Varimax with Kaiser normalization rotation, was used to define the structure of the measured space…“ needed to be rewritten, rephrased and corrected. Also what mathematical procedures??? „…was transformed into a proportional numerical distribution score on a linear scale, where the value of centroid test score for each individual participant was transformed into a proportional point score.
Moreover, this is not the way one should present results of EFA. Rotated component matrix???
Author Response
Dear reviewer 1,
We appreciated your time and effort dedicated to providing feedback to our manuscript and we are grateful for the insightful comments on and valuable improvements to our paper. We have followed all your suggestions and changes are highlighted in track changes in the re-submitted files.
In the attached document, we are going to answer your questions in the same order you send them to us.
Thank you for your valuable work.
Kind regards,
Andrea Perazzetti

Reviewer 2 Report
Comments and Suggestions for Authors
Dear authors,
First of all, I would like to congratulate you on the excellent work carried out. I consider this a high-quality study, conducted with scientific rigor, and providing sport-specific information of great relevance. In the following lines, I will provide some comments on each section of the manuscript.
Abstract
It may be important and relevant to include the number of participants in the study, as this represents a significant strength of the work. Additionally, it would be advisable to describe the results in slightly more detail.
Introduction
The introduction is well written, clearly presenting previous studies on the topic and the analysis of the sport within the scientific literature. The necessity of the present study, as well as its utility and applicability within the field, are well justified.
Materials and Methods
This section is very well developed, precise, and clearly describes each of the key components. In the participants subsection, it might be helpful to expand the sample description in order to justify the classification of the athletes as elite, or even consider some categorization based on their level of performance.
Results
The results are appropriately described, and all analyzed data are reported. However, it may be somewhat difficult for the reader to follow the flow of the results. It would be useful to provide a more detailed description of the data presented in the tables, or even include a figure illustrating the swimming score models across countries, which could help make the results more visually accessible.
Discussion
Some points in the discussion rely too heavily on assumptions, which slightly detracts from the main objective of the study. It would be beneficial to focus more specifically on the analysis of sprint performance in young elite water polo players. Additionally, the structure of this section is somewhat complex and may be challenging for readers to follow; therefore, revising the writing style to ensure greater clarity and coherence is recommended.
Conclusion
The conclusions section effectively synthesizes the main findings of the study and provides very relevant information.
Overall, this is a solid piece of work, and I commend the authors for it. I believe the content and the information provided are highly valuable for the target population described. Addressing some formatting and writing issues would greatly improve the manuscript’s readability.
Kind regards
Author Response
Dear reviewer 2,
We appreciated your time and effort dedicated to providing feedback to our manuscript and we are grateful for the insightful comments on and valuable improvements to our paper. We have followed all your suggestions and changes are highlighted in track changes in the re-submitted files.
In the attached document, we are going to answer your questions in the same order you send them to us.
Thank you for your valuable work,
Kind regards,
Andrea Perazzetti
